# PGI: Pareto-Guided Gradient Interference for Generalized Category Discovery

## Abstract

Generalized Category Discovery (GCD) is a task that concentrates on identifying both base and novel categories in an unlabeled dataset while preserving knowledge from a labeled dataset. A key challenge in this setting lies in balancing the supervised learning of base categories and the unsupervised discovery of novel ones. Through empirical analysis, we observe that conventional multi-objective optimization approaches suffer from significant gradient interference between the classification objective and the representation objective, which hinders effective joint training. Therefore, we propose a simple yet effective framework, named **P**areto-guided **G**radient **I**nterference (**PGI**), to alleviate this issue. The PGI employs a Pareto-annealing optimization approach to explore the Pareto front that balances representation objective and classification objective. Additionally, a regularization term is introduced which can leverage multi-view consistency to enhance clustering structure in the feature space, facilitating better separation of novel classes. Extensive experiments across fine-grained benchmarks demonstrate the superiority of our approach in discovering novel categories while maintaining accuracy on base classes.

## 1 Introduction

Deep neural networks have achieved remarkable success in image classification tasks, largely attributed to the large-scale annotated training datasets. However, this closed-world assumption where all test categories are seen during training poses significant limitations for real-world applications, where numerous unseen and unlabeled categories commonly exist (Wang et al., 2025b). Generalized Category Discovery (GCD) which is a new research branch of Open-set Recognition (OSR), aims to enable the model to classify the base and novel categories from an unlabeled dataset using knowledge from a labeled dataset. GCD is a challenging task that requires models to possess both knowledge transfer capabilities and robust generalization, making it more aligned with real-world scenarios.

Recent studies (Fei et al., 2022; Wu et al., 2023; Cao et al., 2024; An et al., 2024; Shi et al., 2024; Ma et al., 2025) make great progress on coarse-grained datasets, while they remain suboptimal on fine-grained datasets. We hypothesize that one underlying reason for the suboptimal performance of conventional methods on fine-grained datasets may lie in a key issue: **gradient interference among multiple loss functions**. Gradient interference refers to the phenomenon in multi-objective or multi-task learning where gradients from different loss functions conflict with each other—typically indicated by a negative cosine similarity—thereby impeding effective optimization and model convergence (Yu et al., 2020). Fine-grained datasets are characterized by subtle inter-class differences, which often lead to overlapping feature distributions. This increases the possibility of gradient interference when optimizing multiple objectives simultaneously. We divide the loss functions in the GCD task into the classification objective and the representation objective. The classification objective comprises cross-entropy loss and the representation learning objective includes distillation loss, contrastive loss, and various regularization terms. The feasibility analysis of this loss partitioning criterion will be introduced in the experiment section. As shown in Figure 1, our empirical observations reveal significant interference among the gradients between the classification objective and the representation objective, resulting in unstable training dynamics. This observation provides an explanation for the overfitting phenomenon observed in later training stages. In contrast, our

proposed method effectively mitigates gradient interference, leading to more stable and generalized model performance.

To address the challenge of gradient interference, we introduce a principled framework termed **P**areto-Guided **G**radient **I**nterference (PGI). The primary goal of PGI is to dynamically reconcile the optimization between the classification objective, which emphasizes accurate recognition of base categories, and the representation objective, which focuses on learning transferable embeddings that are beneficial for discovering novel categories. Traditional approaches often rely on the static weighting schemes to combine these objectives, which can easily cause one objective to dominate and lead to unstable training dynamics. By contrast, PGI reformulates the problem into a multi-objective optimization task and seeks an update direction that satisfies the Pareto optimality principle. To achieve this, PGI constructs a gradient update direction through a *Pareto-annealing optimization strat-*

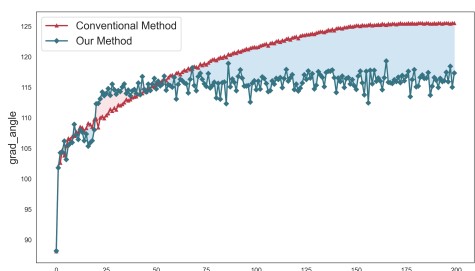

Figure 1: Visualization of gradient interference between conventional method and our proposed method. The conventional method is implemented based on the MOS framework (Peng et al., 2025). The y-axis denotes the gradient angle, which measures the angle between the gradient directions of the classification objective and the representation objective.

*egy*, in which the preference coefficient is gradually annealed during training to reflect the evolving optimization landscape. This dynamic trade-off ensures that both objectives contribute meaningfully at different training stages, allowing the model to explore diverse update directions early on while converging toward Pareto-optimal solutions in later phases. Such a design guarantees that the combined gradient remains a valid descent direction even in the presence of negative cosine similarity, thereby effectively mitigating destructive gradient conflicts. Overall, PGI provides a theoretically grounded and empirically validated approach for alleviating gradient interference, stabilizing convergence, and improving the balance between memorization and generalization in generalized category discovery tasks.

Although Pareto-guided gradient integration reduces the macroscopic conflicts between classification and representation objective, residual interference remains in fine-grained scenarios where subtle semantic differences play a decisive role. To further refine the learned feature space and alleviate these residual conflicts, we incorporate a *multi-view consistency regularization* based on the Swapping Assignments between Views (SwAV) loss (Caron et al., 2020). This component enforces invariance across different augmented views of the same instance by assigning embeddings to a set of prototypes and promoting consistent assignments between views. The SwAV loss encourages predicting the assignment of one view from the similarity distribution of another, thereby forcing the network to capture fine-grained, view-invariant representations. This process not only strengthens the clustering structure within the embedding space but also reduces the tendency of prototypes to overfit to base categories. By embedding SwAV regularization into PGI, the model gains improved sensitivity to subtle category differences while maintaining balanced supervision from both labeled and unlabeled data. Thus, the regularization component complements the Pareto-annealing optimization by refining microscopic structures of representation learning, ultimately enhancing robustness and improving the generalization of the prototype network on challenging fine-grained benchmarks.

To summarize, We provide a comprehensive empirical analysis that reveals the presence of gradient interference among multiple loss functions in generalized category discovery. Then, we propose the **P**areto-Guided **G**radient **I**nterference **PGI** framework, which unifies two complementary components: (i) a Pareto-annealing optimization strategy that dynamically integrates gradients of classification objective and representation objective to approximate Pareto-optimal update directions, and (ii) a SwAV-based multi-view consistency regularization that enhances prototype robustness and promotes fine-grained feature learning. Extensive experiments on multiple fine-grained benchmarks are conducted to demonstrate the effectiveness and generalizability of the proposed PGI framework compared to baselines.

## 2 RELATED WORKS

**Open-Set Recognition.** Open-Set Recognition (OSR) is a realistic open-set task in which the model needs to identify whether or not a test sample belongs to one of the semantic classes in a classifier's training set (Vaze et al., 2021). A standard baseline OSR involves training a model on base classes using cross-entropy loss, with inference decisions based on the maximum value of the softmax probability. Novel Category Discovery (NCD) which is based on OSR and is a more difficult task, aims to cluster an unlabeled set with the help of a labeled set consisting of disjoint but related classes(Wang et al., 2024b). However, OSR focuses solely on identifying test-time samples that do not belong to any of the labeled classes, without requiring further classification of these out-of-distribution instances. NCD typically relies on the restrictive assumption that all unlabeled samples originate from novel classes, an assumption that often does not hold in realistic scenarios. In contrast, GCD addresses this limitation by automatically identifying both base and novel categories from unlabeled data (Rastegar et al., 2023).

**Generalized Category Discovery.** GCD is a challenging semi-supervised learning task which aims to classify both base and novel classes in a labeled dataset, using the knowledge from a unlabel dataset (Vaze et al., 2022). SimGCD (Wen et al., 2023) introduced the parametric classification to replace the conventional methods and gained tremendous improvement on GCD benchmarks. SelEX (Rastegar et al., 2024) proposed self-expertise which enhances the model's ability to recognize subtle differences and uncover unknown categories by using hierarchical pseudo-labeling. SPT-NET (Wang et al., 2024a) introduced a two-stage adapation approach to iteratively optimizes model parameters and data parameters. MOS (Peng et al., 2025) used the object segmentation model to form scene-agnostic images and constrain the model to capture the scene information, achieving prominent improvement in GCD benchmarks. However, many traditional methods ignore the gradient interference between the classification objective and representation objective which hinder the learning efficiency of the model.

**Gradient Interference.** Gradient interference constitutes a fundamental challenge in Multi-Task Learning (MTL), arising when task-specific gradients exhibit divergent directions and heterogeneous magnitudes. This necessitates the design of update strategies that effectively reconcile such conflicts while preserving balanced optimization across tasks (Qin et al., 2025). Recent studies have demonstrated substantial performance improvements through a variety of approaches, including Pareto optimization (Sener & Koltun, 2018), gradient normalization (Chen et al., 2018), conflict projection (Yu et al., 2020), gradient sign dropout (Chen et al., 2020), conflict-averse gradients (Liu et al., 2021), and fair resource allocation (Ban & Ji, 2024).

## 3 METHOD

### 3.1 PRELIMINARIES

**Formal Definition of GCD.** GCD task aims to assign class labels to all unlabeled images, even when their categories may be unseen in the labeled set. Formally, given a set of labeled images $D_L = \{(x_i, y_i)\}_{i=1}^n$ and a set of unlabeled images $D_U = \{x_i\}_{i=n+1}^N$, the goal is to assign class labels to all images in $D_U$. The category of the unlabeled datasets $C_U$ comprises both base and novel classes while the category of the labeled datasets $C_L$ comprises only known classes. Consequently, the category of the labeled datasets $C_L$ is a subset of the category of the unlabeled datasets $C_U$, denoted as $C_L \subset C_U$. It is noted that GCD is a generalization of the traditional category discovery task and is different from the Novel Category Discovery (NCD) task (Han et al., 2019). In NCD, the category of the unlabeled datasets $C_U$ comprises only novel classes, i.e., $C_U \cap C_L = \emptyset$.

**Formal Definition of Gradient Interference.** The gradient interference refers to the phenomenon where the gradients arising from different objectives, tasks, or data subsets in multi-task or hybrid training settings are not aligned in direction, leading to conflicts during parameter updates. For instance, let $\mathcal{L}_i$ and $\mathcal{L}_j$ denote two different objective functions, and the gradient interference occurs when the cosine similarity of their gradients is negative:

$$\cos(g_i, g_j) = \frac{\langle g_i, g_j \rangle}{\|g_i\| \|g_j\|} < 0, \tag{1}$$

where $g_i$ is the gradient of the $\mathcal{L}_i$ loss and $g_j$ is the gradient of the $\mathcal{L}_j$. $\|\cdot\|$ is the norm of gradient and $cos(g_i, g_j)$ is the cosine similarity function between $g_i$ and $g_j$. The gradient interference is formally defined as the set of gradient interactions where cosine similarity is negative.

## 3.2 PARETO-GUIDED GRADIENT INTERFERENCE

Our proposed method **P**areto-guided **G**radient **I**nterference (PGI) is a simple but effective way to balance the classification objective and the representation objective. PGI consists of the pareto annealing optimization technique and multi-view consistency loss.

**Pareto Annealing Optimization.** From the optimization perspective, the multi-task learning can be casted as a multi-objective optimization, with the overall objective of finding a Pareto optimal (Sener & Koltun, 2018). Pareto optimization theory serves as a heuristic mechanism to implicitly steer the model toward a Pareto-optimal solution by aligning task gradients through a normalized, weighted combination. The gradients of the classification objective can be defined as $\nabla_\theta \mathcal{L}_s$, where $\theta$ is the parameters of the model and $\mathcal{L}_s$ denotes the classification objective. Meanwhile, the gradients of the representation objective can be defined as $\nabla_\theta \mathcal{L}_u$, where $\mathcal{L}_u$ denotes the representation objective. The formulation of the representation objective $\mathcal{L}_u$ can be defined as $\mathcal{L}_u = \sum_{j=1}^{|J|} \mathcal{L}_j$, where $\mathcal{L}_j$ means a single representation objective. Then, we can construct the combined gradient update direction $\tilde{\mathcal{G}}_{combined}$ and it can be defined as:

$$\tilde{\mathcal{G}}_{combined} = \alpha \cdot \nabla_\theta \mathcal{L}_s + (1 - \alpha) \cdot \nabla_\theta \mathcal{L}_u, \tag{2}$$

where $\alpha$ is the parameter for balancing the weight of classification objective and representation objective. A static choice of $\alpha$ fixes the trade-off throughout training and may prevent the optimizer from adapting to the evolving curvature landscape of the objectives. To address this limitation, we introduce an annealing scheme for $\alpha$, and the annealing strategy is defined as:

$$\alpha_t = \alpha_{min} + 0.5(\alpha_{max} - \alpha_{min})(1 + cos(\frac{\pi t}{T})), \tag{3}$$

where $\alpha_t$ decreases from $\alpha_{max}$ to $\alpha_{min}$ over the course of T training steps. The previous research MGDA (Désidéri, 2012) demonstrates that any gradient combination constructed as a convex combination with non-negative coefficients summing to one corresponds to a valid descent direction, enabling the approximation of solutions on the Pareto front within the objective space. Therefore, **PGI** is capable of achieving Pareto-optimal solutions and the gradient update formulation is defined as follows:

$$\theta \longleftarrow \theta - \eta \cdot \tilde{\mathcal{G}}_{combined}, \tag{4}$$

where $\eta$ is the learning rate.

**Multi-View Consistency for the Representation objective.** In the aforementioned section, we introduced the pareto annealing optimization technique to balance the gradient update directions between the classification objective and the representation objective. However, a more subtle challenge remained: we found that the model exhibited limited sensitivity to the nuanced, fine-grained distinctions that are essential for many GCD benchmarks. These subtle cues, which are critical for learning robust and discriminative representations, were still being obscured by residual gradient interference. To address this limitation, we incorporate the Swapping Assignments between Views (SwAV) loss (Caron et al., 2020) as a regularization term, encouraging the model to focus on invariant fine-grained features across multiple views.

The formulation of the SwAV loss is defined as follows. Given a pair of images, data augmentation is applied to generate two distinct views $(\mathcal{X}_{view1}, \mathcal{X}_{view2})$. We feed each view into the neural network to obtain logits, formulated as:

$$logit_{view1} = f(\mathcal{X}_{view1}), \tag{5}$$

$$logit_{view2} = f(\mathcal{X}_{view2}), \tag{6}$$

where $f(\cdot)$ represents the neural network. We introduce a sample-level supervision mask $\text{Mask}_{lbl} \in \{0, 1\}^B$ that indicates labeled samples (1) and unlabeled samples (0). Importantly, $\text{Mask}_{lbl}$ is not applied to the input images; it is used only to weight the SwAV loss terms so that contributions are computed on labeled samples. To preserve the structural representation of previously learned classes in the prototype network, we incorporate the Sinkhorn algorithm to enable soft clustering

with balanced assignment constraints (Caron et al., 2020). The Sinkhorn algorithm operates on a sample–prototype similarity matrix, which is constructed using dot products between image embeddings and prototype vectors. It then iteratively normalizes this matrix into a doubly stochastic form, enabling balanced and probabilistic assignments via entropy-regularized optimal transport.

The formulation of sample-prototype similarity matrix $\mathcal{Q}$ is defined as follows:

$$\mathcal{Q}_{view1} = exp(\frac{logit_{view1}}{\tau} - Max(\frac{logit_{view1}}{\tau}))^T \in \mathbb{R}^{K \times B}, \tag{7}$$

$$\mathcal{Q}_{view2} = exp(\frac{logit_{view2}}{\tau} - Max(\frac{logit_{view2}}{\tau}))^T \in \mathbb{R}^{K \times B}, \tag{8}$$

where $K$ is the number of prototypes, $B$ is the batch size and $\max(\frac{logit_{view}}{\tau})$ is used to ensure numerical stability and prevent overflow in softmax computation. After that, all the elements of the matrix are first normalized. Subsequently, the Sinkhorn algorithm is applied, which iteratively normalizes the rows and columns of the matrix in an alternating manner. The pseudo code of the Sinkhorn algorithm is listed in the Appendix A.1. This process yields a clustering assignment matrix $\mathcal{P}$ that not only preserves the original similarity structure but also encourages a roughly balanced assignment of samples across prototypes. The process is formulated as :

$$\mathcal{P}_{view1} = Sinkhorn(\mathcal{Q}_{view1}). \tag{9}$$
$$\mathcal{P}_{view2} = Sinkhorn(\mathcal{Q}_{view2}). \tag{10}$$

We follow the previous work (Caron et al., 2020) and adopt a multi-view interaction prediction approach to enhance representation consistency across different views. The SwAV loss $\mathcal{L}_{swav}$ is defined as:

$$\mathcal{L}_{swav} = -\mathbb{E}\big[\sum \text{Mask}_{\text{lbl}} \left( \mathcal{Q}_{view1} \log(\mathcal{P}_{view2}) + \mathcal{Q}_{view2} \log(\mathcal{P}_{view1})\right)\big]. \tag{11}$$

**Overall losses.** The overall loss function $\mathcal{L}_{overall}$ is composed of the classification objective $\mathcal{L}_s$ and the representation objective $\mathcal{L}_u$. The representation objective $\mathcal{L}_u$ consists of the distillation loss $\mathcal{L}_{cluster}$ , the supervised contrastive loss $\mathcal{L}_{sup\_cl}$, the contrastive loss $\mathcal{L}_{cl}$ and the SwAV loss $\mathcal{L}_{swav}$. The representation objective $\mathcal{L}_u$ is defined as follows:

$$\mathcal{L}_u = \lambda \cdot (\mathcal{L}_{sup\_cl} + \mathcal{L}_{swav}) + (1 - \lambda) \cdot (\mathcal{L}_{cl} + \mathcal{L}_{cluster}), \tag{12}$$

where $\lambda$ is the hyper-parameter. Then, the overall loss $\mathcal{L}_{overall}$ is defined as follows:

$$\mathcal{L}_{overall} = \lambda_{cls} \cdot \mathcal{L}_s + \mathcal{L}_u, \tag{13}$$

where $\lambda_{cls}$ denotes the weighting coefficient that governs the trade-off between the classification objective $\mathcal{L}sup$ and the representation objective $\mathcal{L}u$.

### 3.3 THEORETICAL ANALYSIS OF PGI.

The Pareto-Guided Interference (PGI) framework provides a principled mechanism to mitigate gradient interference between the classification objective and the representation objective. The central idea of PGI is to replace naive aggregation of gradients with a controllable convex combination, parameterized by a preference coefficient $\alpha$. This adjustment guarantees not only the feasibility of simultaneous descent when the gradients are negative cosine similarity but also enables the systematic steering of the steady-state gradient angle toward the metric-optimal region.

**Theorem 3.1 (Feasibility of Simultaneous Descent under Negative Cosine).** Let $g_s = \nabla L_s$ and $g_u = \nabla L_u$ denote the gradients of classification objective and representation objective, with norms $a = \|g_s\|$, $b = \|g_u\|$, and cosine similarity $c = \langle e_s, e_u \rangle < 0$. Consider the Pareto-guided update $g^{(\alpha)} = (1 - \alpha)g_u + \alpha g_s$. For any sufficiently small step size $\eta > 0$, there exists a non-empty open interval of preference coefficients $\alpha$ such that both losses strictly decrease:

$$\Delta L_s < 0, \quad \Delta L_u < 0. \tag{14}$$

This theorem establishes that whenever gradient interference occurs ($c < 0$), one can always identify a nontrivial range of preference weights $\alpha$ to ensure simultaneous descent of both tasks. PGI provides a constructive escape from destructive interference zones where naive gradient descent would fail. The whole proof of theorem 3.1 is provided in Appendix A.3.

**Theorem 3.2 (From Optimal Angle to Metric Improvement).** Let $c = \cos\theta$ denote the cosine similarity between the classification objective and representation objective gradients, and suppose the metric of interest $\mathcal{U}(c)$ is smooth and admits a unique local maximizer at $c = \hat{c}$ with $\mathcal{U}'(\hat{c}) = 0$, $\mathcal{U}''(\hat{c}) < 0$. Denote by $c_\alpha^\star$ the stable equilibrium angle attained under the Pareto-guided dynamics for a fixed $\alpha$. If $c_{\alpha_0}^\star \neq \hat{c}$, then there exists another preference $\alpha_1$ such that:

$$| c_{\alpha_1}^\star - \hat{c} | < | c_{\alpha_0}^\star - \hat{c} | \quad \Rightarrow \quad \mathcal{U}(c_{\alpha_1}^\star) > \mathcal{U}(c_{\alpha_0}^\star). \tag{15}$$

This theorem demonstrates that the preference coefficient $\alpha$ serves as a controllable knob to steer the stable equilibrium angle toward the metric-optimal angle $\hat{c}$. Consequently, PGI not only guarantees feasibility of joint descent but also provably enhances downstream performance by aligning the long-run gradient geometry with metric-optimal conditions. The whole proof of theorem 3.2 is provided in Appendix A.3.

## 4 EXPERIMENTAL RESULTS

### 4.1 EXPERIMENTAL SET-UP

**Datasets.** For the convenience of comparison, we assess our approach in the various fine-grained benchmarks for GCD task. We employ the CUB dataset (Welinder et al., 2010), Stanford Cars dataset (Krause et al., 2013), Herbarium 19 (Tan et al., 2019) and FGVC dataset (Maji et al., 2013). It is also noted that all the datasets are equipped with the Semantic Shift Benchmark (SSB) (Vaze et al., 2021). Following the previous work (Pu et al., 2023), we subsample 50 % of samples as the labeled set $D_L$ with the remaining samples constituting the unlabeled dataset $D_U$.

**Metrics.** In line with the previous research (Peng et al., 2025), we assess the model performance with clustering accuracy (ACC). This metric is defined as follows:

$$ACC = \frac{1}{|D_U|} \sum_{i=1}^{|D_U|} 1(y_i = \tilde{y}_i), \tag{16}$$

where $y_i$ is the ground truth labels and $\tilde{y}_i$ is the predicted labels. To

Table 1: Datasets Partitioning: Labeled and Unlabeled Split.

|  | Label | | UnLabel | |
| --- | --- | --- | --- | --- |
|  | Num | Class | Num | Class |
| CUB | 1.5k | 100 | 4.5k | 200 |
| Stanford Cars | 2.0k | 98 | 6.1k | 196 |
| FGVC-Aircraft | 1.7k | 50 | 5.0k | 50 |
| Herbarium | 8.8K | 341 | 25K | 683 |

comprehensively evaluate the performance of the algorithm, we employ the base accuracy (base acc) to assess its classification capability on base classes, and the novel accuracy (novel acc) to measure its performance on novel classes. In addition, we compute the all accuracy (all acc) as a weighted combination of base acc and novel acc to provide an overall assessment of the algorithm's effectiveness.

**Implementation Details.** We propose the PGI framework, which is a plug-and-play technique. It is noted that all the ablation studies are based on the MOS (Peng et al., 2025) framework. The data augmentation and parameter learning strategies are adopted from prior work (Vaze et al., 2022), ensuring consistency and comparability in performance evaluation. For training, we use a batch size of 128 over 200 epochs, with an initial learning rate of 0.1. The gradient update coefficient $\alpha$ is set to 0.35, the loss weighting factor $\lambda_{cls}$ is 0.35, and the temperature parameter $\tau$ is 0.07. All the experiments are conducted on a single NVIDIA GeForce RTX 4090 GPU.

### 4.2 RESULTS ANALYSIS

The results in Table 2 clearly highlight the effectiveness of incorporating Pareto-Guided Interference (PGI) into different GCD frameworks, namely SimGCD Wen et al. (2023), MOS Peng et al. (2025), and GET Wang et al. (2025a), across three fine-grained classification benchmarks.

For SimGCD, the integration of PGI produces substantial improvements, with overall accuracy rising from 60.3% to 65.0% on CUB, 53.8% to 55.7% on Stanford Cars, and 54.2% to 56.0% on FGVC-Aircraft, leading to an average gain of 2.8%. These gains are consistently observed on both base and novel classes, confirming that PGI enables SimGCD to better preserve prior knowledge

Table 2: Evaluation on the fine-grained datasets. ∗ denotes the reproduced results.

| Methods | CUB | | | Stanford Cars | | | FGVC-Aircraft | | | Avg. | | |
|---|---|---|---|---|---|---|---|---|---|---|---|---|
| | All | Base | Novel | All | Base | Novel | All | Base | Novel | All | Base | Novel |
| GCD | 51.3 | 56.6 | 48.7 | 39 | 57.6 | 29.9 | 45 | 41.1 | 46.9 | 45.1 | 51.8 | 41.8 |
| XCon | 52.1 | 54.3 | 51 | 40.5 | 58.8 | 31.7 | 47.7 | 44.4 | 49.4 | 46.8 | 52.5 | 44 |
| ORCA | 36.3 | 43.8 | 32.6 | 31.9 | 42.2 | 26.9 | 31.6 | 32 | 31.4 | 33.3 | 39.3 | 30.3 |
| DCCL | 63.5 | 60.8 | 64.9 | 43.1 | 55.7 | 36.2 | – | – | – | – | – | – |
| GPC | 52 | 55.5 | 47.5 | 38.2 | 58.9 | 27.4 | 43.3 | 40.7 | 44.8 | 44.5 | 51.7 | 39.9 |
| PIM | 62.7 | 75.7 | 56.2 | 43.1 | 66.9 | 31.6 | – | – | – | – | – | – |
| PromptCAL | 62.9 | 64.4 | 62.1 | 50.2 | 70.1 | 40.6 | 52.2 | 52.2 | 52.3 | 55.1 | 62.2 | 51.7 |
| uGCD | 65.7 | 68 | 64.6 | 56.5 | 68.1 | 50.9 | 53.8 | 55.4 | 53 | 58.7 | 63.8 | 56.2 |
| GCA | **68.8** | 73.4 | **66.6** | 54.4 | 72.1 | 45.8 | 52 | 57.1 | 49.5 | 58.4 | 67.5 | 54 |
| SPTNET | 65.8 | 68.8 | 65.1 | 59 | 79.2 | 49.3 | 59.3 | 61.8 | 58.1 | 61.4 | 69.9 | 57.5 |
| LeGCD | 63.8 | 71.9 | 59.8 | 57.3 | 75.7 | 48.4 | 55 | 61.5 | 51.7 | 58.7 | 69.7 | 53.3 |
| protoGCD | 63.2 | 68.5 | 60.5 | 53.8 | 73.7 | 44.2 | 56.8 | 62.5 | 53.9 | 57.9 | 68.2 | 52.9 |
| SimGCD∗ | 60.3 | 65.6 | 57.7 | 53.8 | 71.9 | 45 | 54.2 | 59.1 | 51.8 | 56.1 | 65.5 | 51.5 |
| SimGCD+PGI | 65.0 | 68.9 | 63.1 | 55.7 | 73.2 | 47.1 | 56.0 | 61.7 | 53.2 | 58.9 | 67.9 | 54.5 |
| MOS∗ | 67.4 | 73.4 | 64.3 | 64.6 | 80.0 | 57.1 | 61.0 | 68.0 | 57.6 | 64.3 | 73.8 | 59.7 |
| MOS+PGI | 68.8 | 75.9 | 65.2 | 65.6 | 82.4 | 57.5 | 62.1 | 69.1 | 58.7 | 65.5 | 75.8 | 60.5 |
| GET∗ | 75.5 | 77.4 | 74.6 | 76.7 | 85.7 | 72.3 | 58.9 | 59.6 | 58.5 | 70.4 | 74.2 | 68.5 |
| GET+PGI | 76.5 | 78.2 | 76.1 | 77.6 | 85.9 | 73.7 | 59.3 | 61.9 | 58.7 | 71.1 | 75.3 | 69.5 |

while facilitating the discovery of novel categories. A similar trend is found when PGI is applied to MOS, where overall accuracy increases to 68.8% on CUB, 65.6% on Stanford Cars, and 62.1% on FGVC-Aircraft, yielding an average improvement of 1.2% compared to the baseline. The performance boost is particularly evident in base classes while maintaining stable generalization on novel categories, underscoring PGI's capacity to balance conflicting optimization goals. Even when combined with the stronger GET framework, PGI continues to yield measurable benefits, with CUB accuracy improving from 75.5% to 76.5%, Stanford Cars from 76.7% to 77.6%, and FGVC-Aircraft from 58.9% to 59.3%, producing a consistent average gain of 0.7%. Taken together, these results confirm that PGI provides reliable improvements across frameworks of varying strength: delivering larger gains for weaker baselines and incremental yet meaningful enhancements for stronger models. This consistent performance demonstrates PGI's scalability and adaptability, validating its role as a general-purpose optimization strategy for generalized category discovery tasks.

Table 3: Results on Herbarium 19 dataset.

| Method | All | Base | Novel |
|---|---|---|---|
| GCD | 35.4 | 51.0 | 27.0 |
| SimGCD | 44.0 | 58.0 | 36.4 |
| GET | 49.7 | 64.5 | 41.7 |
| PGI(ours) | 49.9 | 65.9 | 42.1 |

The results in Table 3 clearly illustrate the effectiveness of integrating our PGI technique into the GET framework for generalized category discovery on Herbarium 19 dataset. Compared to conventional GCD and SimGCD baselines, GET already demonstrates a notable improvement with an overall accuracy of 49.7%, base class accuracy of 64.5%, and novel class accuracy of 41.7%. By further incorporating PGI, the framework achieves superior results, with overall accuracy increasing to 49.9%, base class accuracy improving to 65.9%, and novel class accuracy reaching 42.1%. These consistent gain although modest in absolute value, highlight the ability of PGI to refine gradient optimization and alleviate interference across competing objectives.

Table 4: Comparison of gradient interference under the standard SGD update strategy and the proposed PGI strategy. The gradient interference is measured by the average gradient angle over 200 training epochs.

| | CUB | Stanford Cars | FGVC-Aircraft |
|---|---|---|---|
| Angle | 120.36 | 94.59 | 90.11 |
| Angle(PGI) | **116.69** ↓ | **89.03** ↓ | **85.51** ↓ |

**Gradient Interference Analysis.** To evaluate the effectiveness of PGI in alleviating gradient interference, we conduct a comparative analysis with the standard SGD update strategy. The primary goal of this experiment is to assess whether PGI can better align the gradient directions of the classi-

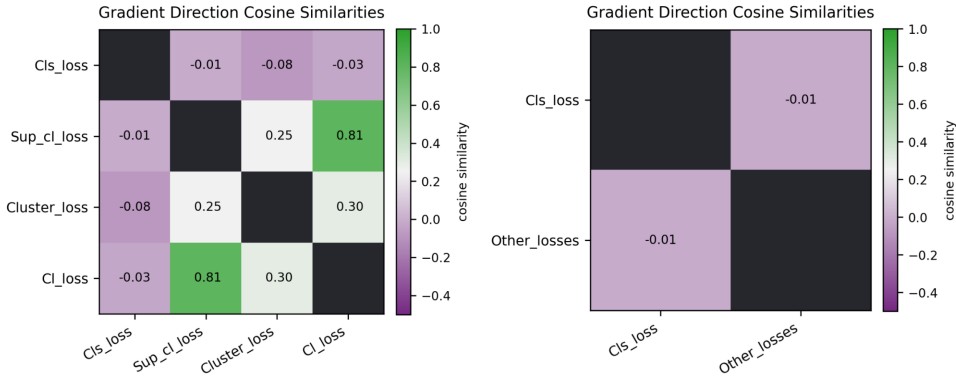

Figure 2: Heatmap of cosine similarity between gradient directions of different loss functions on the CUB dataset. The analysis focuses on the parameters of the last transformer block, where gradient update directions induced by different loss functions are compared to reveal their relationships in GCD.

fication objective and representation objective, thereby reducing their conflict during optimization. As shown in Table 4, PGI consistently reduces the average gradient angle over 200 training epochs across all three datasets: from $120.36°$ to $116.69°$ on CUB, from $94.59°$ to $89.03°$ on Stanford Cars, and from $90.11°$ to $85.51°$ on FGVC-Aircraft. A lower angle reflects improved coherence between gradients, indicating that PGI effectively suppresses interference between learning signals. These results confirm that PGI enhances the optimization stability and facilitates more effective joint training on labeled and unlabeled data.

**Feasibility Analysis of the Loss Partitioning Criterion.** The heatmap in Figure 2 provides clear evidence for the feasibility of partitioning the training objectives into the classification objective and the representation objective. As shown on the left, the cosine similarities among the three representation-related losses (*sup_cl_loss*, *cl_loss*, and *cluster_loss*) are strongly positive, particularly between *sup_cl_loss* and *cl_loss* (0.81), indicating that they share highly aligned gradient directions and jointly promote feature space structuring. In contrast, *cls_loss* exhibits near-zero or negative similarity with each of these losses (e.g., -0.08 with *cluster_loss*), revealing a potential source of gradient interference. When these three representation losses are aggregated into a single objective, as depicted on the right, the correlation structure is preserved, while the antagonism between classification and representation objectives becomes more apparent. This contrast justifies the proposed partitioning: the representation objective forms a coherent group with internal synergy, whereas the classification objective remains orthogonal or conflicting, supporting the rationale for analyzing them separately.

### 4.2.1 ABLATION STUDY.

To validate the improvements achieved by our proposed PGI framework, we conduct extensive ablation studies on the CUB dataset.

**Effectiveness of Each Component.** To comprehensively evaluate the individual contributions of each technical component in our framework, we decompose the overall model into three distinct modules: the baseline, the SwAV loss, and the PGI. Here, the baseline corresponds to the MOS framework, serving as the foundational structure for comparison. As illustrated in Table 5, incorporating only the SwAV loss module yields a 2.7% improvement on novel classes, highlighting its effectiveness in promoting generalization. However, this gain is accompanied by a notable decline in performance on base classes, suggesting that while SwAV loss encourages the learning of transferable features, it may also lead to the erosion of knowledge previously acquired from labeled data. In contrast, integrating only the PGI module results in a 3.2% improvement on base class accuracy, with marginal benefits observed on novel classes. This behavior implies that PGI effectively preserves knowledge related to base categories by guiding gradient updates in a direction that favors retention over exploration. When both modules are combined, the model achieves a favorable trade-off between memorization and generalization, demonstrating superior robustness across class

categories. This synergistic effect confirms the complementary roles of the SwAV loss and the PGI. It is worth noting that, for fair comparison, both modules are trained using a stage-wise strategy when introduced independently, which ensures the stable optimization and avoids the premature convergence.

**Effectiveness of MASK function.** To further investigate the role of supervision in the application of the SwAV loss, we designed a set of ablation studies to compare its effectiveness under different data conditions. Specifically, we examined three settings: applying the SwAV loss only to the labeled data, only to the unlabeled data, and to both simultaneously. As reported in Table 6, the model achieves the best overall performance when the SwAV loss is applied exclusively to the labeled data. This suggests that supervised signals are essential for guiding the clustering-based SwAV mechanism to align feature representations meaningfully with semantic prototypes. In contrast, when the loss is applied solely to the unlabeled data, the model exhibits degraded performance, particularly in the representation of the novel classes. This degradation can be attributed to the lack of explicit semantic anchors, which hampers the structural adaptation of the prototype network and may introduce noise into the optimization process. Furthermore, when SwAV is applied to the entire dataset indiscriminately, the performance improvement becomes marginal. This result indicates that the inclusion of unlabeled data without proper supervision declines the effectiveness of the SwAV's self-supervised objectives. These findings collectively highlight the critical importance of selectively applying SwAV loss to labeled data, where reliable supervision helps stabilize training and enhance prototype discrimination, ultimately leading to better generalization.

Table 5: Comparison of framework components. Baseline refers to the MOS framework. The SwAV Loss refers to the regularization loss. THe PGI refers to the gradient update strategy. All the ablation studies are conducted on the CUB dataset.

| Baseline | SwAV Loss | PGI | CUB | | |
|---|---|---|---|---|---|
| | | | All | Base | Novel |
| ✓ | - | - | 67.4 | 73.6 | 64.3 |
| ✓ | ✓ | - | 68.4 | 71.3 | 67.0 |
| ✓ | - | ✓ | 68.6 | 76.8 | 64.5 |
| ✓ | ✓ | ✓ | **68.8** | **75.9** | **65.2** |

Table 6: Comparison of the different objectives of the MASK function. The W/Labeled indicates that the objective of the mask function is computed on labeled data. The W/Unlabeled indicates that the objective of the mask function is computed on unlabeled data. All the ablation studies are conducted on the CUB dataset.

| W/Labeled | W/Unlabeled | CUB | | |
|---|---|---|---|---|
| | | All | Base | Novel |
| - | - | 68.6 | 76.8 | 64.5 |
| ✓ | - | 68.8 | 75.9 | **65.2** |
| - | ✓ | **69.3** | **79.6** | 64.1 |
| ✓ | ✓ | 67.9 | 74.7 | 64.5 |

## 5 CONCLUSION

In this work, we identify the presence of gradient interference among multiple loss functions in the GCD task, which explains the overfitting phenomenon observed in the later stages of training. To address this issue, we propose the pareto annealing optimization technique, which adaptively fuses the gradient directions of the classification objective and the representation objective to achieve a Pareto-optimal update. Additionally, we incorporate the SwAV loss as a regularization term to enhance the model's sensitivity to fine-grained features and improve the robustness of the prototype network for alleviating the gradient interference. Extensive experiments on fine-grained benchmarks demonstrate the effectiveness and generalization ability of our proposed method. Future research may further explore the theoretical foundations and broader implications of gradient interference in complex learning systems.

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

# A APPENDIX

## A.1 ALGORITHM

The Sinkhorn algorithm presented here performs an entropy-regularized matrix normalization, often used to compute a soft assignment matrix in optimal transport problems or self-supervised learning frameworks. The algorithm begins by rescaling the input matrix out by a temperature factor (here, 0.05), which controls the sharpness of the softmax operation. The maximum value is subtracted from the matrix to improve numerical stability before applying the element-wise exponential function. This transforms the raw scores into a positive matrix Q of shape $K \times B$, where K is the number of prototypes and B is the batch size or number of data samples.

To ensure the matrix represents a valid doubly stochastic assignment (i.e., row sums and column sums meet specific constraints), the algorithm performs three iterations of Sinkhorn normalization. In each iteration, the rows of Q are first normalized so that each prototype (row) has a total mass of 1/K, followed by normalization of the columns so that each sample (column) has a total mass of 1/B. A small constant $\epsilon$ is added to denominators to avoid division by zero. After normalization, the matrix is scaled by B to ensure that the final output sums to 1 along the columns. The final matrix, transposed back to shape $B \times K$, can be interpreted as a soft assignment of each data sample to the K prototype clusters, where each row forms a probability distribution.

---

**Algorithm 1** Sinkhorn Normalization

---

**Require:** Output matrix $O \in \mathbb{R}^{B \times K}$, temperature $\tau = 0.05$, small constant $\varepsilon$
**Ensure:** Normalized assignment matrix $Q \in \mathbb{R}^{B \times K}$

1: $Q \leftarrow \exp\left((O/\tau - \max(O))^T\right)$        // Shape: $K \times B$
2: $Q \leftarrow Q/(\sum Q + \varepsilon)$
3: $K \leftarrow$ number of rows in $Q$        // Number of prototypes
4: $B \leftarrow$ number of columns in $Q$        // Number of samples
5: **for** $i = 1$ to $T$ **do**
6:     $Q \leftarrow Q/(\sum_{\text{rows}} Q + \varepsilon)$
7:     $Q \leftarrow Q/K$
8:     $Q \leftarrow Q/(\sum_{\text{cols}} Q + \varepsilon)$
9:     $Q \leftarrow Q/B$
10: **end for**
11: $Q \leftarrow Q \times B$
12: **return** $Q^T$        // Shape: $B \times K$

---

## A.2 ADDITIONAL EXPERIMENTS

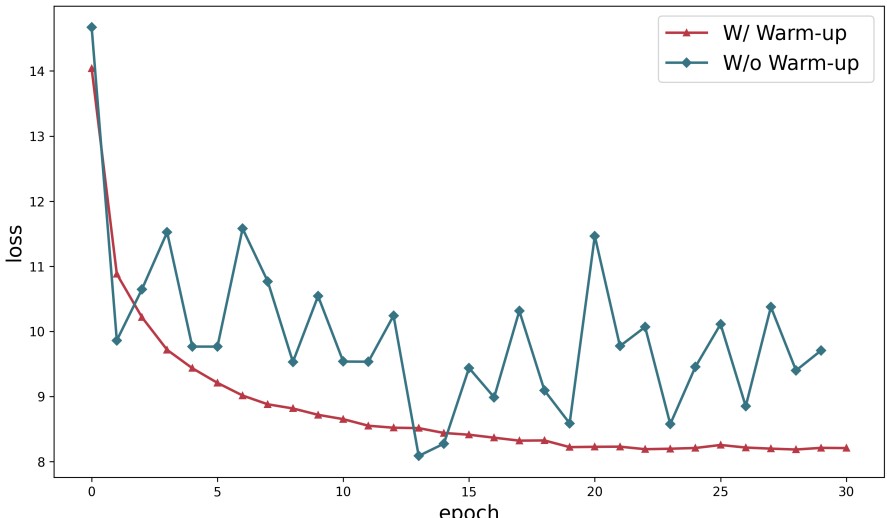

Figure 3: Comparison of loss reduction with and without the warm-up strategy.

It is worth noting that PGI is only employed after the warm-up phase. During the warm-up phase, standard gradient-based optimization is employed to jointly accelerate the model convergence. After the warm-up phase, PGI is applied to gradient updates. This design prevents the model from taking shortcuts by converging to local Pareto-optimal directions that cause training oscillations.

As illustrated in Figure 3, the curve with warm-up strategy shows a smooth and consistent decline in loss, while the curve without warm-up fluctuates significantly and converges less effectively. This indicates that the warm-up strategy stabilizes the early training phase by gradually increasing the learning rate, thereby preventing abrupt updates that may hinder optimization. As a result, the model benefits from improved convergence behavior and reaches a lower and more stable loss, highlighting the effectiveness of warm-up in promoting robust training dynamics.

A.3 PROOF OF THEOREM

**Definition 1** (Loss functions and gradients)**.** *Let $L_s(\theta)$ and $L_u(\theta)$ denote the classification objective and representation objective respectively, where $\theta \in \mathbb{R}^d$ is the vector of trainable parameters. Their gradients are:*

$$g_s := \nabla L_s(\theta), \qquad g_u := \nabla L_u(\theta). \tag{A.1}$$

**Definition 2** (Gradient norms and unit directions)**.** *The norms of the task gradients are:*

$$a := \|g_s\|, \qquad b := \|g_u\|, \tag{A.2}$$

*and the corresponding normalized unit vectors are:*

$$e_s := \frac{g_s}{a}, \qquad e_u := \frac{g_u}{b}. \tag{A.3}$$

**Definition 3** (Cosine similarity)**.** *The cosine similarity between the gradients is:*

$$c := \langle e_s, e_u \rangle = \frac{\langle g_s, g_u \rangle}{ab}, \qquad c \in (-1, 1). \tag{A.4}$$

*We say that gradient interference occurs when $c < 0$.*

**Definition 4** (Pareto-guided update direction)**.** *For a preference parameter $\alpha \in [0, 1]$, the combined gradient is the convex combination:*

$$g^{(\alpha)} := (1 - \alpha)\, g_u + \alpha\, g_s. \tag{A.5}$$

*The parameter update with step size $\eta > 0$ is:*

$$p^+ := p - \eta\, g^{(\alpha)}. \tag{A.6}$$

**Definition 5** (Loss changes after an update)**.** *The first-order changes in the two losses after one update step are:*

$$\Delta L_s := L_s(p^+) - L_s(p), \qquad \Delta L_u := L_u(p^+) - L_u(p). \tag{A.7}$$

**Theorem 3.1 (Feasibility of Simultaneous Descent under Negative Cosine).** Let $g_s = \nabla L_s$ and $g_u = \nabla L_u$ denote the gradients of classification objective and representation objective, with norms $a = \|g_s\|$, $b = \|g_u\|$, and cosine similarity $c = \langle e_s, e_u \rangle < 0$. Consider the Pareto-guided update $g^{(\alpha)} = (1 - \alpha)g_u + \alpha g_s$. For any sufficiently small step size $\eta > 0$, there exists a non-empty open interval of preference coefficients $\alpha$ such that both losses strictly decrease:

$$\Delta L_s < 0, \quad \Delta L_u < 0. \tag{A.8}$$

*Proof.* For any differentiable $L$ and any vector $d$, the first-order Taylor formula at $\theta$ gives:

$$L(\theta + h) = L(\theta) + \langle \nabla L(\theta),\, h \rangle + R_L(h), \qquad \text{with} \quad \frac{R_L(h)}{\|h\|} \xrightarrow[h \to 0]{} 0. \tag{A.9}$$

Equivalently, there exists a constant $K_L \geq 0$ (depending on a neighborhood of $\theta$) such that:

$$|R_L(h)| \leq K_L \|h\|^2 \quad \text{for all sufficiently small } h. \tag{A.10}$$

We apply this with the specific increment $h = -\eta\, g^{(\alpha)}$.

*For $L_s$:*

$$\Delta L_s = L_s\big(p - \eta g^{(\alpha)}\big) - L_s(p) = \underbrace{\langle \nabla L_s(p),\, -\eta g^{(\alpha)} \rangle}_{\text{linear term}} + R_s(-\eta g^{(\alpha)}), \tag{A.11}$$

where $R_s$ satisfies equation A.10. Since $\nabla L_s(p) = g_s$, the linear term equals $-\eta \langle g_s,\, g^{(\alpha)} \rangle$. Hence:

$$\Delta L_s = -\eta \langle g_s, g^{(\alpha)} \rangle + R_s(-\eta g^{(\alpha)}). \tag{A.12}$$

*For $L_u$:* analogously,

$$\Delta L_u = -\eta \langle g_u, g^{(\alpha)} \rangle + R_u(-\eta g^{(\alpha)}), \tag{A.13}$$

with $|R_u(-\eta g^{(\alpha)})| \le K_u \eta^2 \|g^{(\alpha)}\|^2$ for small $\eta$.

Next, we compute the inner products in equation A.12–equation A.13 by *explicitly expanding* $g^{(\alpha)}$:

$$g^{(\alpha)} = (1 - \alpha)g_u + \alpha g_s. \tag{A.14}$$

Therefore,

$$\langle g_s, g^{(\alpha)} \rangle = \langle g_s, (1 - \alpha)g_u + \alpha g_s \rangle = (1 - \alpha)\langle g_s, g_u \rangle + \alpha \langle g_s, g_s \rangle, \tag{A.15}$$

$$\langle g_u, g^{(\alpha)} \rangle = \langle g_u, (1 - \alpha)g_u + \alpha g_s \rangle = (1 - \alpha)\langle g_u, g_u \rangle + \alpha \langle g_u, g_s \rangle. \tag{A.16}$$

By definition of 1, 2, 3,

$$\langle g_s, g_s \rangle = a^2, \qquad \langle g_u, g_u \rangle = b^2, \qquad \langle g_s, g_u \rangle = \langle g_u, g_s \rangle = ab\,c. \tag{A.17}$$

Substituting these into equation A.15–equation A.16, we obtain:

$$\langle g_s, g^{(\alpha)} \rangle = (1 - \alpha)\,ab\,c + \alpha\,a^2, \qquad \langle g_u, g^{(\alpha)} \rangle = (1 - \alpha)\,b^2 + \alpha\,ab\,c. \tag{A.18}$$

Plugging these back into equation A.12–equation A.13 yields the *explicit first-order forms with remainders*:

$$\Delta L_s = -\eta\Big(\alpha a^2 + (1 - \alpha)ab\,c\Big) + R_s(-\eta g^{(\alpha)}), \tag{A.19}$$

$$\Delta L_u = -\eta\Big((1 - \alpha)b^2 + \alpha ab\,c\Big) + R_u(-\eta g^{(\alpha)}). \tag{A.20}$$

Write $c = -|c|$ since $c < 0$. Then the linear parts in equation A.19–equation A.20 become:

$$-\eta\Big(\alpha a^2 + (1 - \alpha)ab\,c\Big) = -\eta\Big(\alpha a^2 - (1 - \alpha)ab\,|c|\Big), \tag{A.21}$$

$$-\eta\Big((1 - \alpha)b^2 + \alpha ab\,c\Big) = -\eta\Big((1 - \alpha)b^2 - \alpha ab\,|c|\Big). \tag{A.22}$$

Because the overall minus sign in equation A.21–equation A.22 multiplies the bracketed expressions, strict decrease at first order is guaranteed if and only if the bracketed expressions are strictly positive.

*For $\Delta L_s$:*

$$\alpha a^2 - (1 - \alpha)ab\,|c| > 0 \iff \alpha a^2 > (1 - \alpha)ab\,|c| \iff \alpha a > (1 - \alpha)b\,|c|. \tag{A.23}$$

Divide both sides by $a + b|c| > 0$:

$$\alpha a + \alpha b|c| > b|c| \iff \alpha(a + b|c|) > b|c| \iff \alpha > \frac{b|c|}{a + b|c|}. \tag{A.24}$$

*For $\Delta L_u$:*

$$(1 - \alpha)b^2 - \alpha ab\,|c| > 0 \iff (1 - \alpha)b > \alpha a\,|c| \iff b > \alpha(b + a|c|). \tag{A.25}$$

Divide both sides by $b + a|c| > 0$:

$$\alpha < \frac{b}{b + a|c|}. \tag{A.26}$$

Compute the width:

$$\frac{b}{b + a|c|} - \frac{b|c|}{a + b|c|} = \frac{b(a + b|c|) - b|c|(b + a|c|)}{(b + a|c|)(a + b|c|)} = \frac{ab\,(1 - |c|^2)}{(b + a|c|)(a + b|c|)}. \tag{A.27}$$

Since $|c| < 1$, the numerator $ab(1 - |c|^2) > 0$; the denominator is also positive. Hence the width is positive and the interval is non-empty.

From equation A.10 there exist constants $K_s, K_u \geq 0$ such that, for all sufficiently small $\eta$,

$$R_s(-\eta g^{(\alpha)})| \leq K_s \eta^2 \|g^{(\alpha)}\|^2, \qquad |R_u(-\eta g^{(\alpha)})| \leq K_u \eta^2 \|g^{(\alpha)}\|^2. \tag{A.28}$$

Moreover, by the triangle inequality,

$$\|g^{(\alpha)}\| \leq (1-\alpha)\|g_u\| + \alpha\|g_s\| \leq a + b. \tag{A.29}$$

Fix $\alpha$ in the interval. Define the *linear margins*:

$$m_s := \alpha a^2 - (1-\alpha)ab\,|c| > 0, \qquad m_u := (1-\alpha)b^2 - \alpha ab\,|c| > 0, \tag{A.30}$$

which are strictly positive by Step 3. Then for $\eta$ satisfying:

$$0 < \eta < \min\left\{ \frac{m_s}{2K_s(a+b)^2}, \frac{m_u}{2K_u(a+b)^2} \right\}, \tag{A.31}$$

we have:

$$|R_s(-\eta g^{(\alpha)})| \leq \tfrac{1}{2}\eta\, m_s, \qquad |R_u(-\eta g^{(\alpha)})| \leq \tfrac{1}{2}\eta\, m_u. \tag{A.32}$$

Therefore, using equation A.19–equation A.20 and equation A.21–equation A.22,

$$\Delta L_s \leq -\eta m_s + \tfrac{1}{2}\eta m_s = -\tfrac{1}{2}\eta m_s < 0, \qquad \Delta L_u \leq -\eta m_u + \tfrac{1}{2}\eta m_u = -\tfrac{1}{2}\eta m_u < 0. \tag{A.33}$$

This proves strict descent of both losses for all sufficiently small $\eta$. $\qquad\square$

**Definition 6** (Metric as a function of the cosine). *Let $\mathcal{U} : (-1,1) \to \mathbb{R}$ be a twice continuously differentiable ($C^2$) downstream metric that depends on the cosine $c$ only. Assume $\mathcal{U}$ has a unique local maximizer at $c = \hat{c}$:*

$$\mathcal{U}'(\hat{c}) = 0, \qquad \mathcal{U}''(\hat{c}) < 0. \tag{A.34}$$

**Definition 7** (Equilibrium angle under Pareto-guided dynamics). *Let $f : (-1,1) \times [0,1] \to \mathbb{R}$ be the angle-dynamics drift induced by the Pareto-guided update (e.g., the ODE $\dot{c} = f(c,\alpha)$ obtained from the second-order analysis). For a fixed $\alpha \in [0,1]$, an* equilibrium angle $c_\alpha^\star$ *satisfies:*

$$f(c_\alpha^\star, \alpha) = 0. \tag{A.35}$$

**assumption 1** (Regularity of the equilibrium map). *There exists a nonempty interval $J \subseteq [0,1]$ such that:*

> *(A1) For every $\alpha \in J$, the angle dynamics admits a unique asymptotically stable equilibrium $c_\alpha^\star \in (-1,1)$.*

> *(A2) The mapping $\alpha \mapsto c_\alpha^\star$ is continuous and strictly monotone on $J$.*

**assumption 2** (Local strict concavity neighborhood). *Since $\mathcal{U}''(\hat{c}) < 0$ and $\mathcal{U} \in C^2$, there exist numbers $\delta > 0$ and $m > 0$ such that on the closed interval:*

$$I_\delta := [\hat{c} - \delta, \hat{c} + \delta] \subset (-1,1) \tag{A.36}$$

*we have the uniform curvature bound:*

$$\mathcal{U}''(c) \leq -m < 0, \qquad \forall c \in I_\delta. \tag{A.37}$$

*Moreover, there exists a nonempty sub-interval $J_\delta \subseteq J$ such that:*

$$c_\alpha^\star \in I_\delta, \qquad \forall \alpha \in J_\delta. \tag{A.38}$$

**Theorem 3.2 (From Optimal Angle to Metric Improvement).** Let $c = \cos\theta$ denote the cosine similarity between the classification objective and representation objective gradients, and suppose the metric of interest $\mathcal{U}(c)$ is smooth and admits a unique local maximizer at $c = \hat{c}$ with $\mathcal{U}'(\hat{c}) = 0$, $\mathcal{U}''(\hat{c}) < 0$. Denote by $c_\alpha^\star$ the stable equilibrium angle attained under the Pareto-guided dynamics for a fixed $\alpha$. If $c_{\alpha_0}^\star \neq \hat{c}$, then there exists another preference $\alpha_1$ such that:

$$|c_{\alpha_1}^\star - \hat{c}| < |c_{\alpha_0}^\star - \hat{c}| \quad \Rightarrow \quad \mathcal{U}(c_{\alpha_1}^\star) > \mathcal{U}(c_{\alpha_0}^\star). \tag{A.39}$$

*Proof.* By Assumption 2, there exist $\delta > 0$ and $m > 0$ such that $\mathcal{U}''(c) \leq -m < 0$ for all $c \in I_\delta = [\hat{c} - \delta, \hat{c} + \delta]$. Hence $\mathcal{U}$ is strictly concave on $I_\delta$ and attains its unique maximum at the interior point $c = \hat{c}$.

By Assumption 2, for all $\alpha \in J_\delta \subseteq J$ we have $c_\alpha^\star \in I_\delta$. In particular, $c_{\alpha_0}^\star \in I_\delta$ (since $\alpha_0 \in J_\delta$). We will choose $\alpha_1 \in J_\delta$ later and verify that $c_{\alpha_1}^\star \in I_\delta$ automatically holds.

Because $c_{\alpha_0}^\star \neq \hat{c}$ and the mapping $\alpha \mapsto c_\alpha^\star$ is continuous and *strictly monotone* on $J_\delta$ by Assumption 1(A2), there exists, for any sufficiently small $\varepsilon > 0$, a point $\alpha_1 \in J_\delta$ such that $c_{\alpha_1}^\star$ lies strictly between $c_{\alpha_0}^\star$ and $\hat{c}$. We make this explicit by a two-case construction:

*Case 1:* $c_{\alpha_0}^\star > \hat{c}$. If $\alpha \mapsto c_\alpha^\star$ is strictly *decreasing* on $J_\delta$, choose $\alpha_1 > \alpha_0$ sufficiently close to $\alpha_0$ so that $\hat{c} < c_{\alpha_1}^\star < c_{\alpha_0}^\star$. If it is strictly *increasing*, choose $\alpha_1 < \alpha_0$ sufficiently close to $\alpha_0$ so that again $\hat{c} < c_{\alpha_1}^\star < c_{\alpha_0}^\star$. In both subcases, by continuity and strict monotonicity, such an $\alpha_1$ exists inside $J_\delta$, and hence $c_{\alpha_1}^\star \in I_\delta$ by Assumption 2.

*Case 2:* $c_{\alpha_0}^\star < \hat{c}$. The construction is symmetric. If $c_\alpha^\star$ is strictly decreasing on $J_\delta$, choose $\alpha_1 < \alpha_0$ sufficiently close so that $c_{\alpha_0}^\star < c_{\alpha_1}^\star < \hat{c}$. If it is strictly increasing, choose $\alpha_1 > \alpha_0$ sufficiently close so that $c_{\alpha_0}^\star < c_{\alpha_1}^\star < \hat{c}$. Again, such $\alpha_1 \in J_\delta$ exists and $c_{\alpha_1}^\star \in I_\delta$.

In either case we have established:

$$\left| c_{\alpha_1}^\star - \hat{c} \right| < \left| c_{\alpha_0}^\star - \hat{c} \right|, \tag{A.40}$$

Fix any $c \in I_\delta$. By Taylor's theorem with Lagrange remainder, for some $\xi$ lying between $c$ and $\hat{c}$,

$$\mathcal{U}(c) = \mathcal{U}(\hat{c}) + \mathcal{U}'(\hat{c})(c - \hat{c}) + \frac{1}{2}\mathcal{U}''(\xi)(c - \hat{c})^2. \tag{A.41}$$

Since $\mathcal{U}'(\hat{c}) = 0$ (by maximality at $\hat{c}$) and $\mathcal{U}''(\xi) \leq -m$ for all $\xi \in I_\delta$, it follows that:

$$\mathcal{U}(\hat{c}) - \mathcal{U}(c) = -\frac{1}{2}\mathcal{U}''(\xi)(c - \hat{c})^2 \geq \frac{m}{2}(c - \hat{c})^2, \qquad \forall c \in I_\delta. \tag{A.42}$$

Because $c_{\alpha_0}^\star, c_{\alpha_1}^\star \in I_\delta$, we can apply equation A.42 twice, obtaining:

$$\mathcal{U}(\hat{c}) - \mathcal{U}(c_{\alpha_0}^\star) \geq \frac{m}{2}\left(c_{\alpha_0}^\star - \hat{c}\right)^2, \qquad \mathcal{U}(\hat{c}) - \mathcal{U}(c_{\alpha_1}^\star) \geq \frac{m}{2}\left(c_{\alpha_1}^\star - \hat{c}\right)^2. \tag{A.43}$$

Subtracting the second inequality from the first and using: $\left(c_{\alpha_1}^\star - \hat{c}\right)^2 < \left(c_{\alpha_0}^\star - \hat{c}\right)^2$ (from Step 3) yields:

$$\mathcal{U}(\hat{c}) - \mathcal{U}(c_{\alpha_0}^\star) > \mathcal{U}(\hat{c}) - \mathcal{U}(c_{\alpha_1}^\star), \tag{A.44}$$

which is equivalent to: $\mathcal{U}(c_{\alpha_1}^\star) > \mathcal{U}(c_{\alpha_0}^\star)$, i.e., equation A.39.

We have explicitly constructed $\alpha_1 \in J_\delta$ such that the associated equilibrium angle is closer to $\hat{c}$ and strictly improves the metric value. This completes the proof. $\qquad\square$

