# OpenReview forum: "PGI: Pareto-Guided Gradient Interference for Generalized Category Discovery"
_ICLR.cc/2026/Conference — Submitted to ICLR 2026_

### Official Review · Reviewer_t7ab · 2025-10-28

**Soundness:** 2
**Presentation:** 2
**Contribution:** 3
**Rating:** 2
**Confidence:** 5

**Summary:**

This paper studies the task of Generalized Category Discovery (GCD). The issue lies in the gradient interference between the classification objective and the representation objective. To solve this problem, this work proposes a simple yet effective framework, named Pareto-guided Gradient Interference (PGI). Besides, the paper further proposes a regularization term using multi-view consistency in SwAV to ensure better representations.

**Strengths:**

1. This paper is well-motivated and easy to follow.
2. The proposed method PGI, serves as a plug-and-play component to enhance existing GCD methods consistently.
3. A thorough theoretical analysis has been presented in the appendix to validate the validity of PGI.

**Weaknesses:**

1. The novelty of this paper is somewhat limited. In the multi-view consistency regularization, the paper directly adopts the swap prediction loss in SwAV. The method seems to just combine existing algorithms in other domains with GCD methods.
2. There are some typos and flaws in the writing and presentation. For example, Appendix A.3 in line 269, line 281 should be correctly referenced. Some typos include L_sup and L_u in Line 151, as well as the method name $\mu$GCD and ProtoGCD in Table 2. Each method in Table 2 should also be cited. The overall writing should be refined.
3. The effect of the hyper-parameters in Eq. 3 should be explored, including $\alpha_{min}, \alpha_{max}$ and T.
4. Why, and to what extent, does gradient intervention affect the performance of GCD? Further in-depth experiments should be conducted.

**Questions:**

Please see weakness. The motivation and influence of the gradient interference on the final performance should be further presented.

---

### Official Review · Reviewer_P9on · 2025-11-01

**Soundness:** 2
**Presentation:** 3
**Contribution:** 3
**Rating:** 4
**Confidence:** 5

**Summary:**

This paper presents a Pareto annealing optimization method that adaptively fuses the gradient directions of the classification and representation objectives to achieve Pareto-optimal updates. In addition, a SwAV loss is introduced to enhance the model's sensitivity to fine-grained features. Experimental results demonstrate that the proposed method outperforms existing approaches in most cases.

**Strengths:**

1.	The motivation of the paper is clear, and the content is easy to follow.
2.	The idea of fusing gradient directions from different objectives is interesting.
3.	The experimental results show that the proposed plug-and-play technique is effective across different methods.

**Weaknesses:**

1.	The ablation study is insufficient, as it is conducted on only one dataset and one method.
2.	The generalization experiments involving different benchmark methods are limited, and more validation is needed.
3.	The performance of MOS and GET is significantly lower than the results reported in their original papers. The reason for this discrepancy remains unclear.
4.	The authors are suggested to further justify the presence of gradient inference with additional experiments, which would help support the rationale behind the motivation.

**Questions:**

See Weaknesses

---

### Official Review · Reviewer_Fzy6 · 2025-11-01

**Soundness:** 2
**Presentation:** 2
**Contribution:** 1
**Rating:** 2
**Confidence:** 5

**Summary:**

The paper targets generalized category discovery (GCD) and claims that one underexplored reason for sub-optimal performance on fine-grained datasets is gradient interference between (i) a classification objective and (ii) a representation objective. To alleviate this, the authors propose PGI (Pareto-Guided Gradient Interference), which

1. combines the two gradients as a convex combination with a cosine-style annealed preference, and
2. adds a SwAV-style multi-view consistency regularization to “refine” the representation side.

They show consistent but small improvements on CUB, Stanford Cars, FGVC-Aircraft, and Herbarium19, and they report that the average gradient angle between the two objectives is slightly reduced when using PGI.

**Strengths:**

1. **Problem framing is reasonable.** The paper clearly separates GCD training signals into a classification part and a representation part, and uses gradient-angle statistics to show these two can conflict, especially on fine-grained datasets. This is an intuitive diagnostic for GCD.
2. **Method is simple and pluggable.** The proposed PGI step is just a convex combination of two gradients with a smooth schedule; it can be inserted into existing GCD pipelines with minimal code changes.
3. **Results are consistent across several fine-grained benchmarks.** Even though the gains are modest, they appear on CUB, Cars, Aircraft, and Herbarium19, which suggests the idea is not overfitted to one dataset.

**Weaknesses:**

1. **“Pareto” is overstated.** The actual update is a hand-designed convex combination with a cosine schedule. This is not what the  PCGrad[1] / CAGrad[2] / MGDA[3] literature calls “exploring the Pareto front”: there is no per-step solving of a Multi-Objective Optimization subproblem, no projection, no dominance test, no preference elicitation, no actual front visualization. It is simply a *hand-scheduled loss-weighting in gradient space*. The paper cites MGDA and says “PGI is capable of achieving Pareto-optimal solutions” but does not show that any iterate is on or near the front. That is an **over-interpretation of existing theory**. In Sec. 4.1 the coefficient is even fixed ($ \alpha = 0.35$) without specifying $\alpha_{min}$ and $\alpha_{max}$ . This effectively collapses the annealing scheme to a fixed weight, so the claimed “Pareto annealing” is not actually evaluated. Please clarify the actual values of $\alpha_{min}$ and $\alpha_{max}$, or report results with the intended schedule.

2. **Motivation is under-validated.** The paper claims gradient interference is a key reason for suboptimal GCD, but the measured angle reductions are small in Table 4 and there is no comparison to standard anti-conflict baselines (PCGrad[1], CAGrad[2], GradNorm[4]), so we cannot tell if PGI is the best way to address it.

3. **Two levels of weighting are used but not justified.**
The paper first defines a gradient-space combination with $\alpha$ (Eq. 2), then defines a loss-space combination with $\lambda_{\text{cls}}$ (Eq. 13):
  * gradient step: $\widetilde{G} = \alpha \nabla L_s + (1 - \alpha)\nabla L_u$
  * loss itself: $L_{\text{overall}} = \lambda_{\text{cls}} L_s + L_u$

This immediately raises the question: which quantity actually controls the trade-off --- $\alpha$ or $\lambda_{\text{cls}}$ --- and why are both needed? If one simply backpropagates $L_{\text{overall}}$, the resulting gradient is already a fixed convex combination of the two components. Adding an additional convex combination on top (PGI) appears redundant unless the authors explain why loss-level weighting alone cannot produce the observed angle changes. This omission makes the method look more like extra manual tuning than a principled optimizer.

4. **SwAV is weakly tied to the main narrative.** The paper sells PGI as “reducing gradient interference,” but when SwAV is added, it is not shown to reduce gradient angles or resolve conflicts; it functions more like a stability bonus than a component derived from the proposed interference analysis.

5. **Gains are within typical run-to-run noise.** Many improvements are ≈0.5–1.0% and some (e.g., Herbarium19) are 0.2%. Without multi-seed results and fully unified implementations of baselines, it is hard to attribute the gains purely to PGI.

6. **Theory is mostly a restatement.** Theorems essentially say “a convex combination of two conflicting gradients can still be a descent direction,” which is standard in MTL, but they do not explain why the specific cosine schedule is necessary or optimal.

7. **No hyperparameter analysis.** The paper does not study sensitivity to $\alpha_{\min}, \alpha_{\max}, T$ or to $\lambda_{\text{cls}}$. Since the method’s effect is precisely to rebalance two conflicting objectives, the lack of such analysis makes it unclear whether the reported gains are robust or just the result of a good setting on these datasets.

[1] Gradient Surgery for Multi-Task Learning

[2] Conflict-Averse Gradient Descent for Multi-Task Learning

[3] Multiple-gradient descent algorithm (mgda) for multiobjective optimization

[4] GradNorm: Gradient Normalization for Adaptive Loss Balancing in Deep Multitask Networks

**Questions:**

1. **Did all reported experiments actually use the cosine schedule for $\alpha$, or was $\alpha$ fixed to 0.35 as in Sec. 4.1?** Please clarify the exact setting used for Tables 2–6.
2. **Why not compare to PCGrad[1] / CAGrad[2] / GradNorm[3]?** These are the closest baselines for “gradient interference,” and without them it is difficult to judge the contribution.
3. **Can you report gradient-angle statistics *after* adding SwAV?** Right now SwAV is justified as “further reducing interference,” but no evidence is shown.
4. **How sensitive is PGI to $\alpha_{\min}, \alpha_{\max}, T, \lambda_{\text{cls}}$?** If the method works only in a narrow range, then the “plug-and-play” claim is weaker.
5. **Please report mean ± std over ≥3 runs for all main results.** This is important because many gains are within 0.5–1.0%.

[1] Gradient Surgery for Multi-Task Learning

[2] Conflict-Averse Gradient Descent for Multi-Task Learning

[3] GradNorm: Gradient Normalization for Adaptive Loss Balancing in Deep Multitask Networks

---

### Official Review · Reviewer_xLLx · 2025-11-02

**Soundness:** 2
**Presentation:** 2
**Contribution:** 2
**Rating:** 4
**Confidence:** 4

**Summary:**

The paper proposes a learning method for fine-grained generalized category discovery (GCD) that introduces two main components: 1) a Pareto-annealing optimization strategy designed to balance the objectives of representation learning and classification by mitigating gradient interference, and  2) a multi-view consistency regularization based on the SwAV loss to enhance feature alignment. The method is evaluated on four fine-grained benchmarks, with comparisons to multiple existing baselines.

**Strengths:**

- The idea of employing a Pareto-annealing optimization scheme to reduce gradient interference between learning objectives is interesting and potentially relevant to the GCD setting.

- The experimental results on fine-grained GCD benchmarks show consistent, albeit modest, improvements over baseline methods.

**Weaknesses:**

- Unclear motivation and formulation. The rationale for decomposing the GCD objective into separate representation learning and classification components, and for treating their gradient interference as the main issue, is not well justified. There are multiple valid ways to structure the GCD loss, and it is not evident that gradient interference is a major source of overfitting. Moreover, the claim that this phenomenon “explains overfitting” (line 053) is not sufficiently supported. In practice, gradients from these objectives may provide complementary rather than conflicting information.

- Limited conceptual contribution. The proposed method combines two existing ideas—Pareto-annealing optimization and SwAV-based regularization—without a clear conceptual or theoretical connection between them. The integration appears somewhat ad hoc, and the contribution is incremental relative to prior literature. Additionally, the focus on fine-grained GCD narrows the method’s general applicability.

- Weak motivation for the annealing strategy. The derivation and intuition behind the annealing formulation in Equation (3) are insufficiently explained. Theoretical analysis of the proposed Pareto Gradient Interference (PGI) framework appears generic and lacks direct linkage to the specific challenges of GCD.

- Insufficient experimental validation. 1) The performance gains reported in Tables 2 and 3 are marginal relative to the baselines. 2) The ablation results (Table 5) show that the contribution of the SwAV loss is inconsistent, raising questions about its necessity. 3) The method is only evaluated on fine-grained datasets, without results on coarse-grained GCD benchmarks, which limits the generality of the claims.

**Questions:**

- Can the authors clarify why gradient interference is particularly problematic in GCD, and how it theoretically or empirically leads to overfitting?

- What is the underlying motivation or derivation behind the annealing function in Equation (3)?

- Could the authors provide additional results on coarse-grained GCD datasets to demonstrate broader applicability?

---

### Meta-Review · Area_Chair_1tad · 2026-01-05

**Summary:**

The paper initially got consistently negative scores: 4, 4, 2, 2. The main weaknesses raised by the reviewers are: unclear motivation, limited novelty, insufficient experiments and the presentation should be improved. The authors did not provide a rebuttal. Thus the AC believes the reviewers will not change their original scores and recommends rejection to this paper.

**Reviewer Concerns:**

The authors did not provide a rebuttal. Thus all the concerns are remained.

**Reviewer Scores:**

The authors did not provide a rebuttal. Thus the reviewers will not change their scores.

---

### Decision · Program_Chairs · 2026-01-26

Reject